# Molecular Mechanism of Microbiota Metabolites in Preterm Birth: Pathological and Therapeutic Insights

**DOI:** 10.3390/ijms22158145

**Published:** 2021-07-29

**Authors:** AbuZar Ansari, Shambhunath Bose, Youngah You, Sunwha Park, Youngju Kim

**Affiliations:** 1Department of Obstetrics and Gynecology, Ewha Medical Research Institute, College of Medicine, Ewha Womans University, Mokdong Hospital, Seoul 07985, Korea; abu.kim.0313@gmail.com (A.A.); yerang02@naver.com (Y.Y.); clarrissa15@gmail.com (S.P.); 2Department of Bioscience, Sri Sathya Sai University for Human Excellence, Navanihal, Okali Post, Kamalapur, Kalaburagi, Karnataka 585313, India; sambhbasu@gmail.com

**Keywords:** preterm, microbiota, metabolites, molecular mechanism, probiotics, postbiotics

## Abstract

Preterm birth (PTB) refers to the birth of infants before 37 weeks of gestation and is a challenging issue worldwide. Evidence reveals that PTB is a multifactorial dysregulation mediated by a complex molecular mechanism. Thus, a better understanding of the complex molecular mechanisms underlying PTB is a prerequisite to explore effective therapeutic approaches. During early pregnancy, various physiological and metabolic changes occur as a result of endocrine and immune metabolism. The microbiota controls the physiological and metabolic mechanism of the host homeostasis, and dysbiosis of maternal microbial homeostasis dysregulates the mechanistic of fetal developmental processes and directly affects the birth outcome. Accumulating evidence indicates that metabolic dysregulation in the maternal or fetal membranes stimulates the inflammatory cytokines, which may positively progress the PTB. Although labour is regarded as an inflammatory process, it is still unclear how microbial dysbiosis could regulate the molecular mechanism of PTB. In this review based on recent research, we focused on both the pathological and therapeutic contribution of microbiota-generated metabolites to PTB and the possible molecular mechanisms.

## 1. Introduction

Preterm birth (PTB) refers to the birth of infants before 37 weeks of gestation by World Health Organization [1]. PTB is a challenging issue worldwide with a prevalence of 5% to 18%, which increases the risk of morbidity and mortality or long-term complications to neonatal life [2,3]. During early pregnancy, various physiological and metabolic changes occur as a result of endocrine and immune metabolism [4]. Environmental and clinical factors such as toxicant particulate matter (PM 2.5–10, bisphenol, etc.), consumption of a high-fat diet, family PTB history, low education, low socioeconomic status, ethnicity (PTB is higher in non-Hispanic black women), previous PTB history, multiple pregnancies, short pregnancy interval, early (<16 years) or late (>36 years) pregnancy, tobacco or alcohol consumption, high stress, hypertension, obesity, low body mass index, infection, short cervix, uterine anomaly, and miscarriage can affect fetal developmental plasticity, gestational age, or birth outcome [5,6,7,8,9].

Based on clinical observations, PTB is classified as spontaneous PTB (sPTB) and iatrogenic; sPTB is due to preterm labour (PTL) or preterm premature rupture of the membranes (P-PROM) [10,11]. Several lines of evidence indicate that sPTB is commonly associated with intrauterine infection/inflammation [12,13]. Increased inflammatory molecules in different maternal bio-fluids indicate the onset of PTB [13,14,15]. It has been noted that inflammatory molecules (IL1, TNF, and IL6) are associated with the initiation of PTB and as predictive markers of PTB in symptomatic women [14]. Moreover, polymorphisms or hyper-methylation in genes or RNA transcript expression mediates inflammation and are associated with PTB [16,17,18].

Accumulating evidence indicates that the host microbiota regulates the maternal and fetal immune interaction as well as the birth outcome [19,20,21]. In addition, current lines of evidence also indicate that the host microbiota-generated metabolites control various metabolic mechanisms and inflammatory processes including PTB (Figure 1) [22,23]. Although labour is regarded as an inflammatory process, it is still unclear which microbiota and metabolites control PTB. In this review, we focused on the recent research-based evidence to elucidate the probable molecular basis of the involvement of microbial metabolites in PTB, emphasizing both pathogenic and therapeutic insights.

## 2. Microbiota Pathological Insight in PTB

In early pregnancy, various physiological processes change dynamically, including hormonal and immunity for placentation and implementation [24,25]. During the course of normal pregnancy, healthy microbiota colonization can be a prerequisite for immune maturation as well as metabolic and hormonal homeostasis [26,27,28]. However, during pregnancy, the microbiota remains relatively stable or fluctuates dramatically in different trimesters at different regions [29,30,31,32]. Concerning healthy pregnancies, the intrauterine cavity microbiota colonization originates exclusively from ascending route through the urogenital tract (urinary, cervical, and vaginal) and the hematogenous route through the placenta after translocation from the digestive tract (oral and gut) [33]. The oral and gut microbiota stability is affected by extrinsic factors, especially diets, which influence the cervicovaginal microbiota dynamics [34,35]. The vaginal microbiota fluctuates almost daily because of its unstable environment in pregnant women [36]. Normally, the dominance of *Lactobacillus* spp. in the vaginal tract reflects healthy microbial colonization as members of this bacterial spp. promote the maintenance of vaginal homeostasis and prohibit the colonization and growth of adverse microorganisms, including those contributing to sexually transmitted infections. The protective role of *Lactobacillus* spp. is exerted through several mechanisms, such as the creation of an acidic environment by reducing vaginal pH at around 4.0, the production of bioactive compounds, competition for nutrients and adhesion sites, and modulation of host immunity [37,38,39,40]. Instead of the defensive function of dominant *Lactobacillus* spp. and low level of host-derived antimicrobial peptide, immune modulator β-Defensin 2 in cervicovaginal fluid is associated with increased risk of PTB in African American women [41].

A previous study analyzing microbial species composition in 396 samples from a population of asymptomatic North American women representing four ethnic groups (white, black, Hispanic, and Asian) by pyrosequencing of barcoded 16S rRNA genes revealed the clustering of vaginal microbial taxonomic communities into five groups called community state types (CST) [42]. Among them, four were dominated by *Lactobacillus* spp. as follows: CST I (*Lactobacillus crispatus*), CST II (*Lactobacillus gasseri*), CST III (*Lactobacillus iners*), and CST V (*Lactobacillus jensenii*), while CST IV was represented by a lower proportion of Lactobacillus and an array of strict and facultative anaerobes including *Prevotella, Dialister, Atopobium, Gardnerella, Megasphaera, Peptoniphilus, Sneathia, Eggerthella, Aerococcus, Finegoldia*, and *Mobiluncus*. Additionally, communities in CST I have the lowest median pH (4.0 ± 0.3), whereas communities belonging to CST IV had the highest median pH (5.3 ± 0.6) [42,43]. It was found that women with both CST IV and short cervixes are at a higher risk for sPTB than women with either factor alone [44]. In contrast, in another study, the association between term birth and PTB with *Lactobacillus* community composition classified vaginal microbiota into three categories: normal (>90% *Lactobacillus* spp.), intermediate (30–90% *Lactobacillus* spp.), and dysbiotic (<30% *Lactobacillus* spp.) [30,45], *Gardnerella vaginalis* is commonly isolated from patients with BV, while for characterization by 16S rRNA gene of V2 region, PCR products are an indicator of BV [46,47].

It has been found that in early pregnancy, higher microbial richness and diversity in different bio-fluids (blood, urine, cervicovaginal fluid, amniotic fluid) are similar to non-pregnant women before the second trimester. The microbial dysbiosis occurring during the very crucial second trimester, along with racial disparity, directly affects the normal developmental physiology and birth outcome [30,35]. Broadly speaking, dysbiosis in Gram-negative bacteria acts as an inducer of PTB [48,49]. In early pregnancy, an increase in pathogenic microbiota (*Gardnerella, Ureaplasma, Closteridiam, Provetella, Mycoplasma*) provides permissible colonization and metabolic signatures of bacterial vaginosis (BV), which double the risk of PTB [50,51,52,53]. Additionally, African American women have a higher rate of BV-related microbiota than Caucasian women [54]. The microbial translocation is not yet clear, but the ascending and hematogenous route pathways are common, as mentioned earlier [33]. The abundance of *Lactobacillus* spp., particularly the *Lactobacillus crispatus*-dominated vaginal environment (CST I), maintains gestational health and results in term birth. While the *Lactobacillus iners* dominated vaginal milieu (CST III), and there was an abundance of *Clostridiales, Bacteroidales*, and *Actinomycetales,* which are known to lead to vaginal dysbiosis associated with PTB [31,43,55,56]. In addition, the relatively lower abundance of vaginal *Lactobacillus crispatus* and relatively higher abundance of *Anaerococcus vaginalis* and *Prevotella timonensis* were observed in obese women and are significantly related to BV and PTB [8,30].

The abundance of *Bacteroides* and *Escherichia-Shigella* were observed in the blood, while the dominance of BV-associated *Leptotrichia/Sneathia, Mobiluncus* spp., and *Mycoplasma* spp. was reported in the vagina of PTB women [35,51,57]. A high level of Lactobacillales was observed in the feces and an abundance of *Weissella* and *Rickettsiales* were observed in the blood of women who reached full term [35,58]. The abundance of *Ureaplasma* spp. and the family *Veillonellaceae* (including *Megasphaera* spp.) was observed in the urine of PTB women [59]. The BV strains *Sneathia sanguinegens* and *Fusobacterium nucleatum* were identified in amniotic fluid samples at mid-trimester of women with PTB [60]. The low levels of *Clostridium* subcluster (XVIII, XIVa) and *Bacteroides* and a high level of *Lactobacillales* were observed in the feces of the PTB [58]. In pregnancy, the oral microbiota generally exhibits a relatively stable bacterial population to the vaginal microbiome. However, one study has revealed that high levels of common periodontal pathogens *Porphyromonas gingivalis*, *Tannerella forsythia, Prevotella intermedia,* and *Prevotella nigrescens* are associated with an increased risk of PTB [61]. *Lactobacillus crispatus* and *Lactobacillus iners* have a protective role against pathogenic microbiota such as *Gardnerella, Ureaplasma, Closteridiam,* and *Provetella* through beneficial metabolites to prevent virginal dysbiosis, maintain vaginal pH, and protect mucus layer integrity [62,63]. Furthermore, subjects with high concentrations of *Lactobacillus crispatus* at follow-up had high concentrations of metabolites negatively associated with BV, which affected their pregnancies (Figure 2, Table 1) [38,52,56,64].

## 3. Microbiota Metabolites Pathological Insight in PTB 

Microbiota affects the metabolic process directly or indirectly by their generated metabolites, also termed as ‘post-biotic metabolites’ (PBM), which produce similar or much better effects compared to their own live parents [22,68]. In general, bacterial metabolites may impact human cell function, inflammation, and disease susceptibility. Small molecule metabolites (the metabolome) represent the enzymatic pathways and complex metabolic networks that execute microbial transformation of host-derived products. Various external environmental factors, especially diet containing high carbohydrates, high protein, and high fat affect the gut microbiota dysbiosis and metabolic dysregulation [69,70]. Disturbed metabolic dysregulation due to high consumption of carbohydrate and fat diet increases the chances of obesity and relative abundance of pathogenic microbiota [71,72]. Consumption of high carbohydrates, high-protein, high-fat, and/or high-vitamin diets influences maternal extra and intrauterine factors by their pathogenic microbiota and metabolites, thereby increasing the risk of PTB [23,73]. In connection, microbiota dysbiosis directly affects the production of microbiota-metabolites, and the presence of metabolites at higher or lower levels impacts metabolic function including PTB [74,75]. These PBM are represented by several active compounds including short-chain fatty acids, polyamines, polyphosphates, and peptides, which exert a significant effect on several metabolic activities [76]. Principally, there are two basic phenomena involved in the generation of microbial PBM: first, the bio-production of short-chain fatty acids (SCFAs) or alcohol from fermenting sugars or fibers, and second, the bio-conversion of derivative molecules [77,78]. Microbial metabolites generated from pathogenic bacteria such as peptidoglycan (PGN), lipopolysaccharides (LPS), and lipoteichoic acid (LTA) represent pathogen-associated molecular patterns (PAMPs), while damage-associated molecular patterns (DAMPs) are derived from dietary factors. These PAMPs and DAMPs are generated in response to infection and inflammation [79]. These two microbe-specific molecular signatures are recognized by the innate immune system via germline-encoded pattern-recognition receptors (PRRs). In the mammalian system, among the major members of PRR families, Toll-like receptors (*TLRs*) were the first to be identified, and are the best-characterized molecules. Following PAMPs and DAMPs recognition, *TLRs* recruit Toll/*IL1* receptor (*IL1R*) domain-containing adaptor proteins such as *MyD88* and *TRIF*, which induce signal transduction pathways that ultimately lead to the activation of transcription factors *NFκB* and *IRFs* or *MAP* kinases to regulate the expression of pro-inflammatory cytokines, chemokines, and type I IFNs. Such events dictate the outcome of innate immune responses that protect the host from microbial infections [79].

The SCFAs (formate, acetate, succinate, and lactate) are produced from indigestible carbohydrates and dietary fibers in the presence of microbiota. Formate is produced by *Lactobacillus pentosus*, acetate is produced by *Lactobacillus acidophilus* CRL 1014, while various strains of *Clostridium* spp., *Ruminococcus* spp. produce butyrate, propionate, and succinate. Alcoholic metabolites (methanol, ethanol, formate, and isopropanol) are generated through fermentation-mediated production of methanol and ethylene by ammonia-oxidizing bacteria such as *Nitrosomonas europaea* and *Nitrosococcus oceani* [80,81], while acetone is produced mostly by *Clostridium acetobutylicum, Clostridium beijerinckii*, and *Clostridium saccharobutylicum*. Ethylene glycol is produced by *Corynebacterium glutamicum*, glycolate by *Corynebacterium glutamicum*, isopropanol by *Clostridium acetobutylicum* ATCC 824, and *Escherichia coli,* and ethanol by *Lactobacillus fermentum*. Several bacterial pathogens (LPS) depend on polyamines for their survival and virulence within the host, including *Helicobacter pylori*, *Salmonella enterica* subsp. *enterica* serovar *Typhimurium, Shigella* spp., *Staphylococcus aureus, Streptococcus pneumonia*, and *Vibrio cholera* [82]. Finally, the derived metabolite ‘TMAO’ is a bioconverted derivative of TMA that is generated from choline, betaine, and carnitine by the action of eight distinct bacterial strains, including *Anaerococcus hydrogenalis, Clostridium asparagiforme, Clostridium hathewayi, Clostridium sporogenes, Edwardsiella tarda, Escherichia fergusonii, Proteus penneri*, and *Providencia rettgeri* [83]. *Lactobacillus iners* produces a pore-forming toxin (Inerolysin) similar to the one produced by *Gardenella vaginalis*, which is capable of lysing erythrocytes and increase the pH facilitate PTB (<4.5; Figure 3, Table 2) [84].

The generation or availability of microbial metabolites is directly influenced by the dysbiosis of the microbial population, which in turn is affected by environmental factors, especially diet [36]. During pregnancy, microbiota-generated post-biotic metabolites concentration depends upon microbial richness and diversity in different bio-fluids (blood, urine, cervicovaginal fluid, amniotic fluid), which are associated with PTB (Table 3). It has been reported that the levels of polyamines that are associated with mothers’ dietary intake are higher in the preterm women’s breast milk [107]. Alcoholic and acetone microbial metabolites are harmful to the birth outcome. Exposure to a low level of methanol can shorten the pregnancy and promote labour complications, while ethylene metabolite, i.e., ethylene oxide may increase the risk of spontaneous abortion, PTB, and post-term birth. It has been found that the consumption of natural highly sugar-sweetened (e.g., fructose) or artificially sweetened (e.g., aspartame) beverages may be likened to an increased risk of PTB. Aspartame breaks down into methanol and other substances in the body, which in turn can be converted into toxic metabolites such as formaldehyde and formate that adversely affect PTB [23]. LPS is a microbiota endotoxin, which acts on *TLRs* and induces PTB [108]. The high protein content of the amniotic fluid observed in the second-trimester is also considered as one of the contributing factors for PTB [109] since the high level of protein metabolites TMA/TMAO in the second-trimester are known to be associated with PTB [23].

## 4. Molecular Mechanism of Microbiota Metabolites in PTB

Although labour is considered an inflammatory process, accumulating evidence indicates that the molecular mechanisms underlying PTB are multifactorial, involving many biological pathways [11]. More specifically, the estrogen metabolism pathway, intrauterine infection, extracellular matrix degradation, fetal stress, and fetal anomalies are the most reported pathways associated with PTB [118]. However, the most common mechanism of PTB is found to be linked to the inflammatory signaling pathways [12,118]. Several reports indicate the activation of inflammatory reactions in the gestational tissues and secretion of inflammatory cytokines as an immune response to the ascending infection of the genital tract and pathogenic microbial composition [10,74]. More specifically, microbe-induced inflammatory signals arising from urinary tract infection, sexually transmitted infections, including trichomoniasis, or BV are the major factors contributing to PTB [119,120]. The abundance of certain *Lactobacilli* in the vagina has been shown to trigger a distinct inflammatory cascade that largely contributes to CST-specific response. It was revealed that the vaginal presence of *Lactobacillus iners* in CST III and CST IV were associated with a higher baseline in pro-inflammatory factors including macrophage migration inhibitory factor (MIF), IL1α, IL18, and *TNF*, which are known to induce the activation of inflammatory responses [43,121]. In agreement, a previous clinical study performing longitudinal analyses of 16S rRNA, metagenomic, metatranscriptomic, and nine cytokine profiles from forty-five preterm and ninety term birth controls demonstrated higher vaginal levels of eotaxin, IL1β, IL6, and macrophage inflammatory protein (MIP)1β in PTB compared to TB samples [122]. The study also found a strong negative correlation between *Lactobacillus crispatus* and several taxa associated with dysbiosis and PTB (for instance, *Gardnerella. vaginalis, Prevotella cluster 2, S. amnii,* and, to a lesser extent, TM7-H1), as well as with vaginal cytokines, further supporting the benefits of *Lactobacillus crispatus* on vaginal health [121]. Additionally, the examined cytokines, which were largely pro-inflammatory, showed a loose correlation both with each other and with taxa associated with dysbiosis and PTB. While the proinflammatory chemokine IP10/CXCL10 was positively correlated with *Lactobacillus iners*. In contrast, in women with PTB, the proinflammatory cytokines and dysbiotic taxa (for instance, *Atopobium vaginae, Gardnerella vaginalis*, and *Megasphaera type* 1) exhibited a tighter cluster, signifying a stronger positive correlation [122]. Using 16S rRNA and GC-MS/LC–MS a correlation between microbiota (*Gardnerella vaginalis*), and metabolites (2-hydroxyisovalerate and γ-hydroxybutyrate) was observed and identified the biomarkers for clinical BV [123]. Additionally, shortgun sequencing of vaginal microbiota is a powerful molecular technique that reveals the community profiles, as well as functional potential regarding PTB [124].

In recent years, evidence on maternal interactions with microbial metabolites and associated immune responses contributing to the adverse pregnancy outcomes, including PTB, has emerged [125,126,127]. It has been found that dysbiosis of the vaginal, gut, or placental microbiota and subsequent alterations to secondary metabolite biosynthesis are vital for the onset and progression of infection, inflammation, and pathogenesis of PTL [74,125,127,128]. A previous clinical study revealed significant alterations in lipid metabolism in BV as reflected by the higher levels of 12-hydroxyeicosatetraenoic acid (12-HETE), a signaling eicosanoid mediating inflammatory response pathways, and lower levels of its precursor arachidonate, suggesting bioconversion of arachidonate to 12-HETE by BV-associated microbes [52]. Chorioamnionitis, an inflammatory condition of the fetal membranes (amnion and chorion) usually caused by bacterial infection, is known to contribute to PTB. It has been found that chorioamnionitis membranes are often positive for vaginal organisms, particularly those involved in BV [129]. Alterations in the biosynthesis of secondary metabolites (e.g., phenylpropanoid, stilbenoid, diarylheptanoid, and gingerol) and lipid (glycerolipid, glycerophospholipid, arachidonic acid, and unsaturated fatty acids) metabolism accompanied by a higher abundance of oral commensal bacteria-*Streptococcus thermophilus* and *Fusobacterium* sp. are seen in women with chorioamnionitis [130].

Microbial metabolites functionally play an important role in the proliferation, differentiation, and development of epithelial cells, as well as in the maintenance of homeostasis of the immune system [131]. Reports of untargeted metabolites of microbiota generated in the vaginal fluid (formate, methanol, acetone, and TMAO), blood (retinyl palmitate, At-Retinal, 13-cis-Retinoic acid), and targeted metabolites (folate) in the blood reveal a significant association with inflammation, which facilitate PTB cascades [23,73,132]. PTB has also been found to be associated with T-cell activation, which is involved in adaptive immune response [133]. It has been found that glucose and glucose-derived metabolites regulate T-cell activation and signaling through the modulation of particular pathways. For instance, succinate and fumarate, two important metabolites in both the host and microbial processes, are the potent allosteric inhibitors of the 2OG-dependent dioxygenases, which are the members of histone demethylases [134]. Accumulating evidence indicates that epigenetic events, such as histone modifications including methylations are often associated with T-cell activation, differentiation, and commitment [135]. Therefore, it is conceivable that the production and consumption of these metabolites and their transport from the mitochondria to the cytosol facilitating histone methylation dynamics in the nucleus may contribute to the PTB [134].

As aforementioned, several lines of evidence indicate that BV is associated with PTB [136]. BV is manifested by an alteration in the proportion of a particular bacterial population affecting the profile of metabolites in vaginal fluid accompanied by increased cell-shedding from the cervicovaginal epithelium [137]. More specifically, BV is represented by a shift in the vaginal microbial population from the normally Lactobacillus-dominated to a highly complex polymicrobial community characterized by the presence of anaerobic bacteria, such as *Gardnerella vaginalis, Atopobium* spp., *Prevotella* spp., and high levels of several biogenic amines (putrescine, cadaverine, and trimethylamine), short-chain fatty acids (especially acetate and succinate), and low concentrations of certain amino acids (tyrosine and glutamate) [138,139,140]. These findings are also in agreement with a previous clinical study that used mass spectrometry to link specific metabolites with particular bacteria detected in the human vagina by broad-range PCR [52]. The report demonstrated dramatic differences in metabolite compositions and concentrations associated with BV by addressing a global metabolomics approach. More specifically, a total of 279 named biochemicals were detected; among them, the levels of 62% of metabolites in women with BV were significantly different from those in women without BV. BV was particularly associated with strong metabolic signatures across multiple pathways influencing amino acid, carbohydrate, and lipid metabolism. Furthermore, unsupervised clustering of metabolites separated subjects with BV from participants without BV. More specifically, women with BV had metabolite profiles characterized by lower concentrations of amino acids and dipeptides, accompanied by higher levels of amino acid catabolites. Such events indicate augmented utilization of amino acids and increased catabolism in BV, supporting the notion that BV-associated bacteria may use amino acids as a source of carbon and nitrogen. This is in contrast to lactobacilli, which are known to metabolize sugars, such as glycogen. Furthermore, in agreement with previous reports described above, this study also detected well-known amines putrescine, cadaverine, and tyramine in women with BV. N-acetylputrescine (a degradation product of putrescine), cadaverine, and tyramine were associated with elevated pH. Such BV-specific signatures were found to be associated with the presence and concentrations of particular vaginal bacteria. The study also revealed that BV-associated bacterial levels were positively correlated with succinate, while lactobacilli were negatively associated [52].

In early pregnancy, permissible colonization by BV-associated pathogens induces secretion of pro-inflammatory cytokines in vaginal epithelial cells, and BV doubles the risk of PTB [51]. Studies have revealed that microbial compositions of the cervicovaginal fluid (CVF) are associated with metabolic profiles in healthy pregnancy [30,38] Reduce lactate is associated with BV, while succinate acts as an immunomodulatory molecule [141]. *Lactobacillus* abundance has a strong positive association with lactate and, to a lesser extent, with levels of several amino acids, such as isoleucine, leucine, tryptophan, phenylalanine, and aspartate [142]. Lactic acid generated by *Lactobacillus species, L. crispatus* in particular, acidifies the vaginal environment and thereby strongly prevents the growth of potentially harmful microorganisms [143]. In addition, vitamins play a significant role in the composition of various microbiota, including vitamin A metabolite (retinoic acid) as a key player in embryogenesis, and vitamin D shows an immunomodulatory through *TLRs* pathway and effect pregnancy [144,145]. Deficiency or efficiency of vitamins (A, or D, etc.), significantly reflect the microbiota dysbiosis, which might directly affect the production of their metabolites and immunomodulation [146,147]. During pregnancy, deficiency of vitamin D (25-hydroxy (OH) and 1, 25-dihydroxy (OH) 2), associated with PTB, while increased concentrations of vitamin A metabolites (retinyl palmitate, At-Retinal, 13-cis-Retinoic acid), also contribute to PTB [75,111].

As mentioned before, *TLRs* (*TLR2, TLR3, TLR4, TLR5, TLR6*) play an important role in the inflammatory activation processes by binding to the PAMPs or DAMPs [79]. These molecules serve as upstream mediators of the synthesis of inflammatory cytokines and chemokines in infections/inflammation-induced PTB [148]. The metabolites that belong to PAMPs (endotoxins and exotoxins) activate PRRs such as *TLRs* and nod-like receptors that are expressed by amnion epithelial cells, intermediate trophoblasts in the chorion, decidual cells, macrophages, and neutrophils. LPS, an endotoxin and an essential component of the outer membrane of Gram-negative bacteria, acts on *TLRs*, manifesting a strong response to immune systems and an induction of PTB [108,149]. The *PRRs* induce the pleiotropic NFκB signal transduction pathway which regulates the expression of proinflammatory chemokines (e.g., IL8 and C-C motif ligand 2 (CCL2)), cytokines (e.g., IL1β, IL6, TNFα, IFNγ), prostaglandins, and proteases, leading to activation of the common pathway of parturition [118,125] (Figure 4).

## 5. Microbiota Metabolites Therapeutic Insight in PTB

Specifically, a wide variety of drugs designed to inhibit the contraction of myometrial smooth muscle cells are commonly used to prevent PTB. However, therapies are largely ineffective in delaying the delivery and reducing neonatal mortality substantially [12]. As an alternative, many studies have evaluated the potential of probiotics to restrain PTB as they are known to displace and kill pathogens and modulate the immune response by interfering with the inflammatory cascade that leads to PTL and PTB [150]. Although live lactic acid bacteria (LAB) are commonly used as probiotics to treat a wide range of diseases, they are shown to potentiate PTB [151]. However, novel trends in probiotics supplementation are oriented towards the replacement of live microbes with non-viable microbial extracts and metabolic by-products, the PBM [76]. This new approach reduces health risks associated with the consumption of live bacteria, especially concerning their high immune-stimulating potential [22]. Recent data showed that *Lactobacillus postbiotic* metabolites can modulate inflammatory pathways and have potential cytoprotective effects against hepatotoxicity [152]. Healthy microbiota is associated with maintaining a low pH (4–4.5) of vaginal fluid by lactate-produced *Lactobacilli* and maternal physiological factors. *Lactobacillus crispatus* was shown to dominate, and they have been shown to inhibit the growth of *Escherichia coli* and biofilm formation by *Gardnerella vaginalis* [56]. Antimicrobial and immune-modulatory effects of lactic acid and SCFAs produced by vaginal microbiota associated with microbiota eubiosis and BV [153]. LPS increases fetal membrane expression of GPR43, which was significantly higher in women delivering preterm. GPR43-SCFA interactions may represent novel pathways that regulate inflammatory processes involved in labour. In addition, *Lactobacilli* produce hydrogen peroxide and secrete various factors such as bacteriocins and anti-adhesive molecules that suppress the growth of *Gardnerella vaginalis* and compete for anaerobic species [63]. Lactobacillus GR-1 and RC-14 with metronidazole vaginal gel have been used to treat symptomatic BV [154]. It has been found that the lactic acid generated by *Lactobacillus species*, *Lactobacillus crispatus* in particular, acidifies the vaginal environment and thereby strongly prevents the growth of potentially harmful microorganisms [155]

A healthy dietary pattern containing fibrous food consumption during pregnancy decreases the risk of PTB by increasing beneficial SCFA metabolites and lowering pH, compared with a diet consisting of the consumption of Western-style junk foods [156,157]. Prebiotic and probiotics consumption created a barrier effect and protected against pathogens and metabolites production during the gestational period and birth control [158,159]. In early-life exposure to SCFAs during a critical window, protection against pathogenic microbiota through immunopathologies was revealed [160]. SCFA consists of anti-inflammatory role labour, though modulating inflammatory pathways in fetal membranes through *GPR43* and *GPR41 RAR*-related orphan receptor gamma t–positive (*RORγt* +) [160]. Suppression of inflammatory pathways by SCFA may be therapeutically beneficial for pregnant women at risk of pathogen-induced PTB [161]. Ritodrine is a phenethylamine (amine) derivative by certain bacteria (*Lactobacillus, Clostridium, Pseudomonas,* and the family *Enterobacteriaceae*) and acts as a potent antimicrobial against certain pathogenic strains of *Escherichia coli* [162]. Phenethylamine derivatives isolated from the strain of *Arenibacter nanhaiticus* sp. nov. NH36AT consist of antimicrobial activity against *Staphylococcus aureus* and *Bacillus subtilis,*
*Escherichia coli* [163]. *Lactobacillus iners* also produce a moderate level of lactic acid and prevent BV [164]. These microbiota metabolites consist of therapeutic effects on PTB. Consumption of controlled carbohydrate and protein-rich diets decreases the production of toxic microbial metabolites and reduces the risk of PTB. Identification of microbiota metabolites of pathogenic bacteria could be used as a non-invasive, quick, and cost-effective proxy marker for the characterization of the prevailing microbial community and the attendant inflammatory mechanisms of inflammation-induced PTB, as well as uncover potential novel therapies. More specifically, the replacement of live microbes with beneficial post-biotic metabolites might account for a promising therapeutic option to reduce the risks of PTB. Additionally, adequate amounts of vitamins (A, D) supplements reduced the chances of their pathogenic metabolites production and risk of PTB [75,111,165].

## 6. Conclusions

The interaction of microbial metabolites in the gestational stage is involved in both maternal and neonatal health, and pathogenic metabolites increase the risk of PTB. To understand the molecular mechanisms underlying such events, there is a need to elucidate the role of the microbiota and their metabolites in pregnant women. Comparative analyses of omics markers in maternal and fetal bio-fluid (plasma, cervicovaginal, and amniotic fluid) at different trimesters in a well-defined population could reveal the accurate cellular and molecular mechanisms, predictive biomarkers, and biotics-mediated therapeutic approaches for PTB. The advancement in omics research opens a new horizon to elucidate the precise cellular and molecular events in the mechanistic pathway of PTB.

## Figures and Tables

**Figure 1 ijms-22-08145-f001:**
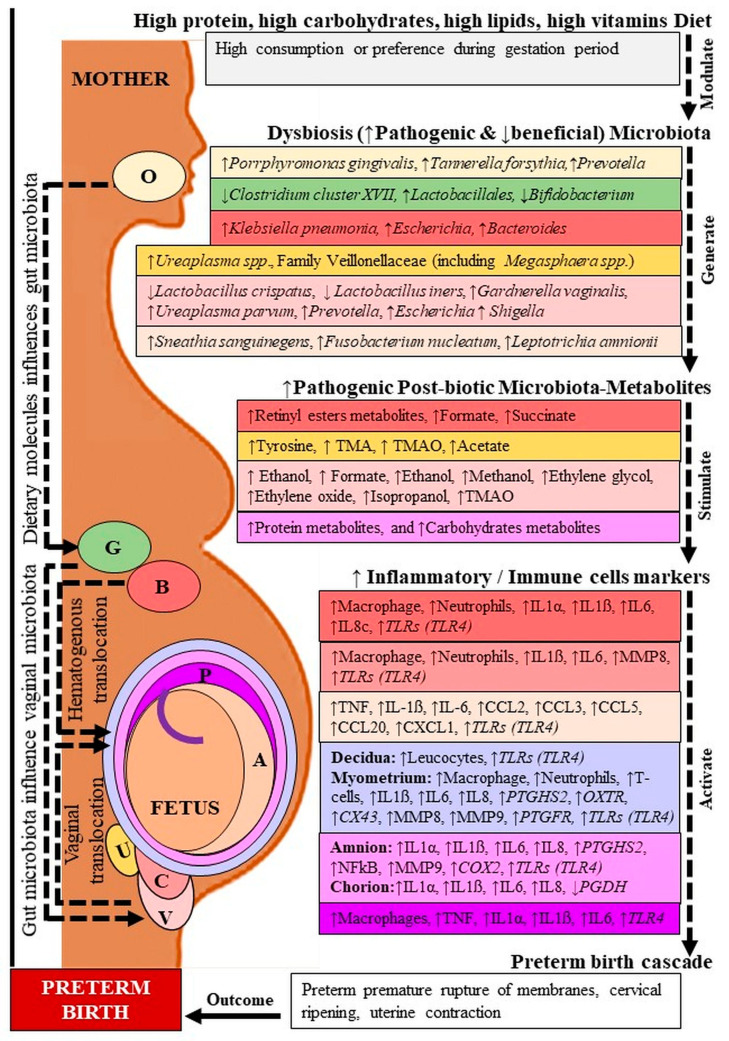
Microbiota-metabolites and inflammatory markers in preterm birth. O: Oral; G: Gut; B: Blood; U: Urine; V: Vagina; C: Cervix; P: Placenta; A: Amniotic fluid. TMA: Trimethylamine, TMAO: Trimethylamine N-oxide; IL: Interleukin; TNF: Tumor necrosis factor; MMP: Matrix metalloproteinase; CCL: C-C motif chemokine ligand; CXCL: C-X-C motif chemokine; *PTGDS2*: Prostaglandin D2 synthase; *OXTR*: Oxytocin receptor; *CX*: Connexin; *NFkB*: Nuclear factor-kappa B; *COX*: Cyclooxygenase; *PGDH*: Prostaglandin dehydrogenase. Dashed lines indicate several steps, ↓ decreased, ↑ increased level. Location followed by color code.

**Figure 2 ijms-22-08145-f002:**
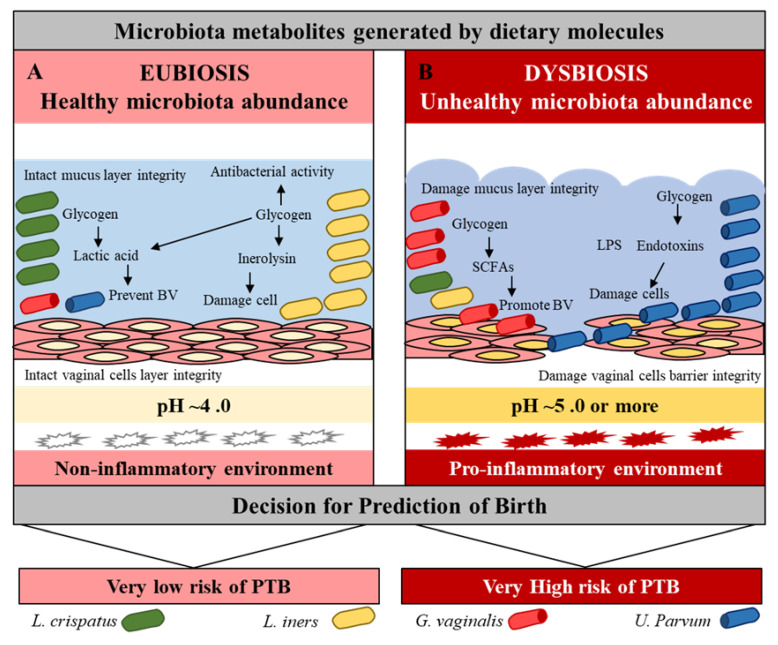
Vaginal microbiota eubiosis and dysbiosis. Effect of beneficial and pathogenic microbiota in the vaginal environment. Pink = normal, while red = inflammation conditions.

**Figure 3 ijms-22-08145-f003:**
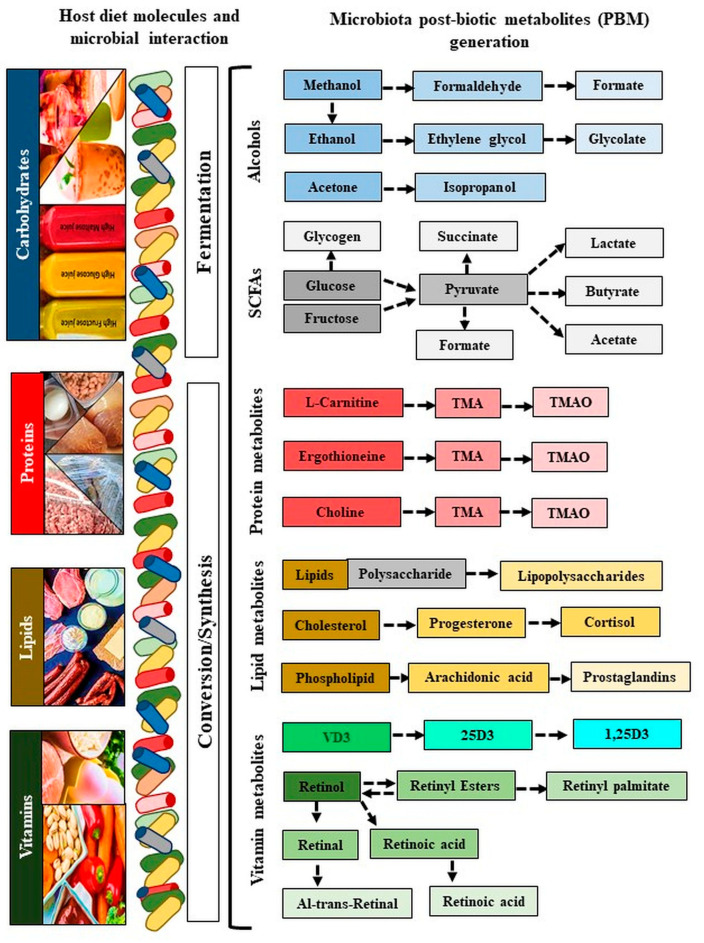
Generation of microbiota metabolites. The blue color refers to carbohydrates, the red color refers to proteins, the yellow color refers to lipids, and the green color refers to vitamins. Colors indicate metabolites generated by their specific diatery molecules and microbiota. Colors from dark to light color shed indicate derivatives of metabolites conversion. VD: Vitamin D; 25D: 25-hydroxy vitamin D; 1,25(OH)2D3: 1α,25-dihydroxy vitamin D.

**Figure 4 ijms-22-08145-f004:**
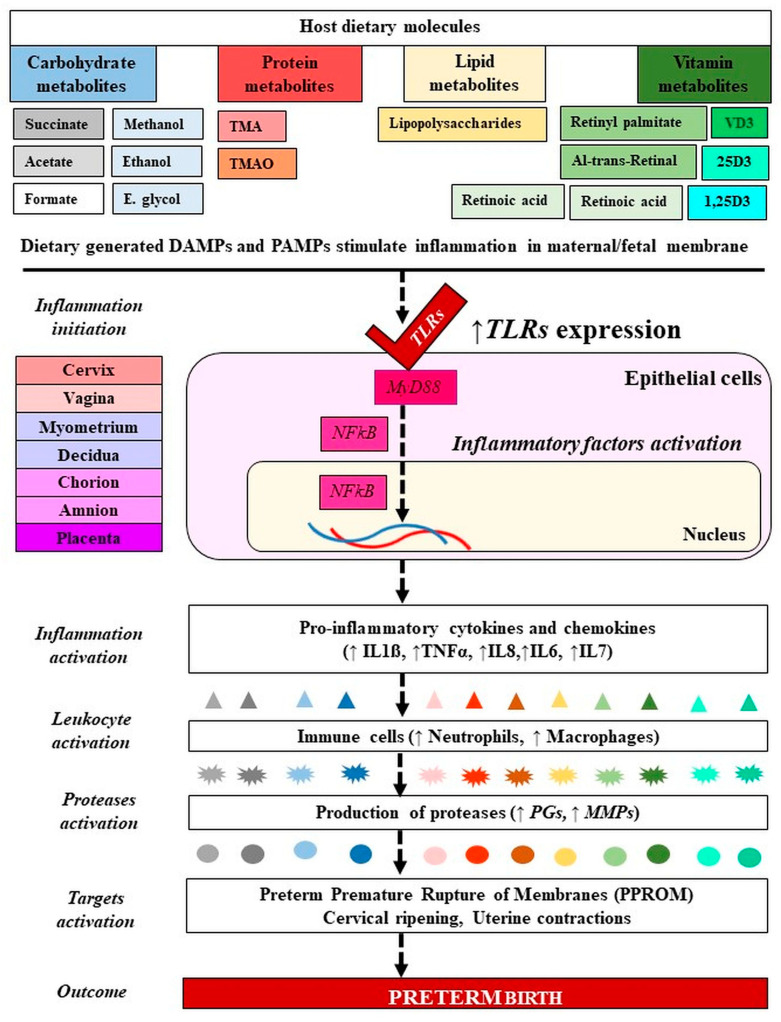
Molecular mechanism of microbiota-metabolites in preterm birth. Signaling of PAMPs and DAMPs derived from gut microbiota and dietary molecules have activated *TLRs* in PTB. The blue and gray colors refer to carbohydrates, the red color refers to proteins, the yellow color refers to lipids, and the green color refers to vitamins. From dark color to light color indicate metabolites forms and their generated inflammatory markers. PAMPs: Ppathogen-associated molecular patterns, DAMPs: Damage-associated molecular patterns; VD: Vitamin D; 25D: 25-hydroxy vitamin D; 1,25(OH)2D3: 1α,25-dihydroxy vitamin D. TMA: Trimethylamine, TMAO: Trimethylamine N-oxide; IL: Interleukin; TNF: Tumor necrosis factor; MMP: Matrix metalloproteinase; PGs: Prostaglandins. *TLRs*: Toll like receptors. Dashed lines indicate several steps, ↓ decreased, ↑ increased level.

**Table 1 ijms-22-08145-t001:** Microbial richness in preterm birth.

Specimen *	Bacterial Strains	References
Oral	*Porphyromonas gingivalis, Tannerella forsythia, Prevotella intermedia,* and *Prevotella nigrescens*	[61]
Blood	*Klebsiella pneumonia*	[65]
	*Bacteroides, Lactobacillus, Sphingomonas, Fastidiosipila, Weissella,* and *Butyricicoccus*	[35]
Vaginal fluid	*Bacteroides, Sphingomonas, Delftia, Lactobacillus crispatus,* and *Escherichia-Shigella*	[30]
	*Prevotella, Lactobacillus, Gardnerella*	[66]
Urine	*Ureaplasma* spp. Family *Veillonellaceae* (including *Megasphaera* spp.)	[59]
Amniotic fluid	*Sneathia sanguinegens, Fusobacterium nucleatum*, and *Leptotrichia amnionii*	[60,67]
Maternal feces	*Lactobacillales*, *Clostridium* cluster XVII, *Clostridium* subcluster XIVa	[58]

* Analyzed in women who gave PTB (1, 2, or 3 trimesters), observed significant differences with term birth.

**Table 2 ijms-22-08145-t002:** Metabolites generate by microbiota.

Microbiota	Metabolites	Chemical Formula	Class	References
*Lactobacillus pentosus*	Formate	CH_2_O_2_	SCFA	[85]
*Prevotella* spp., *Streptococcus* spp. *Bacteroides* spp., *Clostridium* spp., *Ruminococcus* spp. * Blautia hydrogentrophica*, *Bifidobacterium longum* SP 07/3, *Bifidobacterium bifidum* MF 20/5), *Lactobacillus acidophilus* CRL 1014, *Akkermansia muciniphilia*	Acetate	CH_3_COOH	SCFA	[86,87,88,89]
*Ruminococcus* spp., *Salmonella* spp., *Veillonella* spp., *Bacteroides* spp., *Clostridium* spp. *Dalister succinatiphilus, Eubacterium halli, Megasphaera elsdenii, Phascolarctobacterium succinatutens, Roseburia inulinivorans, Akkermansia muciniphilia, Coprococcus catus, Phascolarctobacterium succinatutens, Lactobacillus rhamnosus GG* (LGG), *Lactobacillus gasseri* PA 16/8, *Bifidobacterium longum* SP 07/3, *Bifidobacterium bifidum* MF 20/5), *Lactobacillus salivarius *spp.* salcinius* JCM 1230, *Lactobacillus agilis* JCM 1048, *Lactobacillus acidophilus* CRL 1014	Propionate	CH_3_CH_2_COOH	SCFA	[87,88,90,91]
*Anaerostipes* spp., *Clostridium* spp., *Ruminococcus* spp. *Coprococcus catus, Roseburia inulinivorans, Roseburia intestinalis, Coprococcus comes, Coprococcus eutactus, Clostridium symbiosum, Eubacterium rectale, Eubacterium hallii, Faecalibacterium prausnitzii, Lactobacillus salivarius *spp.* salcinius* JCM 1230, *Lactobacillus agilis* JCM 1048, *Lactobacillus acidophilus* CRL 1014	Butyrate	CH_3_(CH_2_)_2_COOH	SCFA	[90,92,93,94]
*Bifidobacterium* spp., *Lactobacillus* spp. *Bifidobacterium longum* SP 07/3, *Bifidobacterium bifidum* MF 20/5, *Lactobacillus rhamnosus* GG (LGG), *Lactobacillus gasseri* PA 16/8, *Lactobacillus salivarius *spp.* salcinius* JCM 1230, *Lactobacillus agilis* JCM 1048, *Lactobacillus acidophilus* CRL 1014	Lactate	CH_3_CH(OH)CO_2_H	SCFA	[88,95,96]
*Prevotella copri, Ruminococcus flavefaciens, Phascolarctobacterium succinatutens*	Succinate	(CH_2_)_2_(CO_2_H)_2_	SCFA	[97,98]
*Nitrosomonas europaea* and *Nitrosococcus oceani*	Methanol	CH_3_OH	Alcohol	[80,81]
*Lactobacillus fermentum, Weissella confuse,* and *Saccharomyces cerevisiae, Zymomonas mobilis,*	Ethanol	CH_3_CH_2_OH	Alcohol	[99]
*Clostridium acetobutylicum* ATCC 824, *Escherichia coli*	Isopropanol	CH_3_CHOHCH_3_	Alcohol	[100,101]
*Corynebacterium glutamicum*	Glycolate	C_2_H_3_O_3_	Alcohol	[102]
*Clostridium *sp.* strain* G10	Acetone	CH_3_COCH_3_	Alcohol	[103]
*Corynebacterium glutamicum*	Ethylene glycol	(CH_2_OH)_2_	Alcohol	[102]
*Anaerococcus hydrogenalis, Clostridium asparagiforme, Clostridium hathewayi, Clostridium sporogenes, Edwardsiella tarda, Escherichia fergusonii, Proteus penneri* and *Providencia rettgeri*	TMAO	(CH_3_)_3_NO	Protein	[104]
*Helicobacter pylori,* and *Salmonella enterica*	LPS	C_175_H_317_N_5_O_101_P_6_	Lipid	[105]
*Lactobacillus iners*	Inerolysin	INY	Lipid	[84]
*Bifidobacterium*	Folate	C_19_H_19_N_7_O_6_	Vitamin	[106]

Note: SCFA = Short chain fatty acid, TMAO = Trimethylamine N-oxide, LPS = Lipopolysaccharide.

**Table 3 ijms-22-08145-t003:** Microbiota metabolites generated in preterm birth.

Specimen *	Metabolites	References
Blood	Vitamin A metabolites (Retinyl palmitate, At-Retinal, 13-cis-Retinoic acid), Vitamin D metabolites (25-hydroxy (OH) and 1,25-dihydroxy (OH)2 vit D)	[110,111]
Vaginal fluid	Ethanol, Methanol, Ethylene glycol, Ethylene oxide, Isopropanol, TMAO	[23]
Urine	Tyrosine, TMA, TMAO, Acetate, Formate, Phthalate, Choline	[112,113,114,115]
Amniotic fluid	High protein, carbohydrate, and fats	[109]
Maternal feces	Fatty acids and cholesterol hormone metabolites	[116]
Breast milk	Polyamines	[107]
Fetal feces	Acetate and lactate	[117]

* Sample analyzed in PTB (1, 2, or 3 trimesters). Note: TMAO = Trimethylamine N-oxide. Significant difference between term and preterm birth subjects.

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
