# Peer review of "Molecular Mechanism of Microbiota Metabolites in Preterm Birth: Pathological and Therapeutic Insights"

_ijms, 2021, doi:10.3390/ijms22158145_

Round 1

Reviewer 1 Report

Preterm birth is a major contributor to neonatal mortality and long-term morbidity. There are lines of evidences to suggest the association between microbes and preterm birth. In this review, the author reviewed the literature on the pathological and therapeutical contributions of microbiota-generated metabolites to PTB and possible molecular mechanisms.

I recommend this review to be published given this is an important topic in modern obstetrics. I wish to make some comments/suggestions for minor revision:

1 - Lines 122-6: "Microbiota metabolites generated from pathogenic bacteria, such as peptidoglycan, lipopolysaccharides, adn lipoteichoic acid, represent pathogen-associated molecular patterns (PAMPs), while damage-associated molecular patterns (DAMPs) are derived from dietary factors. These PAMPs and DAMPs are generated in response to infection and inflammation (Ref 53, a review on Toll-like receptor)."

Readers of IJMS may not necessarily be familiar with immunology. So, I'd suggest that the authors to write a few more sentences to introduce DAMPs and PAMPs as pattern-recognition receptors (PRRs) for the initial detection of microbes, and also their relation with toll-like receptors. E.g. "Toll-like receptors (TLRs) are pivotal in the recognition of pathogen-associated molecular patterns, such as pathogen-associated molecular patterns (PAMPs), while damage-associated molecular patterns (DAMPs), which are derived from various microbes. TLRs then signal through the recruitment of specific adaptor molecules, leading to activation of the transcription factors NF-κB and IRFs, which dictate the outcome of innate immune responses.

2 - Lines 181-183: "... Evidence of maternal interaction with microbial metabolites and associated immune response contributing to advese pregnancy outcomes, including PTB, is emerging (Ref 59, a review)." 
Lines 293-297: Comparative analyses of omics markers in maternal and fetal biofluid (plasma, ...) at different trimesters ... could reveal ... markers ... for PTB (no citation)."

Systematic Identification of Spontaneous Preterm Birth-Associated RNA Transcripts in Maternal Plasma
https://journals.plos.org/plosone/article?id=10.1371/journal.pone.0034328

In a review, the readers expect to see more citations of original research papers, instead of just another review. Perphaps, you may wish to cite the following paper which use omics techiniques to identify panels of markers in maternal plasma for prediction of preterm birth:

I am sure there are other original reseach papers which you could cite for Lines 181-183, 293-297. So please cite those as well instead of just reviews.

Author Response

Reviewer 1.

Revers reply

Preterm birth is a major contributor to neonatal mortality and long-term morbidity. There are lines of evidences to suggest the association between microbes and preterm birth. In this review, the author reviewed the literature on the pathological and therapeutical contributions of microbiota-generated metabolites to PTB and possible molecular mechanisms.

I recommend this review to be published given this is an important topic in modern obstetrics. I wish to make some comments/suggestions for minor revision:

Reply: Thank you for your insightful response and recommendation for the publication of our manuscript. Following your comments and suggestions, our replies are as follows.

1 - Lines 122-6: "Microbiota metabolites generated from pathogenic bacteria, such as peptidoglycan, lipopolysaccharides, adn lipoteichoic acid, represent pathogen-associated molecular patterns (PAMPs), while damage-associated molecular patterns (DAMPs) are derived from dietary factors. These PAMPs and DAMPs are generated in response to infection and inflammation (Ref 53, a review on Toll-like receptor)."

Readers of IJMS may not necessarily be familiar with immunology. So, I'd suggest that the authors to write a few more sentences to introduce DAMPs and PAMPs as pattern-recognition receptors (PRRs) for the initial detection of microbes, and also their relation with toll-like receptors. E.g. "Toll-like receptors (TLRs) are pivotal in the recognition of pathogen-associated molecular patterns, such as pathogen-associated molecular patterns (PAMPs), while damage-associated molecular patterns (DAMPs), which are derived from various microbes. TLRs then signal through the recruitment of specific adaptor molecules, leading to activation of the transcription factors NF-κB and IRFs, which dictate the outcome of innate immune responses.

Reply: Thank you for this insightful suggestion. In the revised version (Lines 122-126 → Lines 173-177 in the revised version) of our manuscript, we increased the content about the PAMPs and DAMPS regarding immune response along with related references (Lines 176-185 in the revised version; Reference 81).

2 - Lines 181-183: "... Evidence of maternal interaction with microbial metabolites and associated immune response contributing to advese pregnancy outcomes, including PTB, is emerging (Ref 59, a review)." 

Reply: Thank you for this suggestion. In the revised version (Lines 181-183 → Lines 269-271 in revised version) of our manuscript, we incorporated the references of original and current research articles, instead of review articles (References 126-128 in the revised version).

Lines 293-297: Comparative analyses of omics markers in maternal and fetal biofluid (plasma,..) at different trimesters ... could reveal ... markers ... for PTB (no citation)."

Reply: In the revised version (Lines 293-297 → Lines 349-350 in the revised version) of our manuscript, we cited more references regarding the omics technique with PTB markers (References 31, 116, 124, and 140 in the revised version), and because of the related section in the “Conclusion” of our manuscript, we did not cite any references.

Systematic Identification of Spontaneous Preterm Birth-Associated RNA Transcripts in Maternal Plasma
https://journals.plos.org/plosone/article?id=10.1371/journal.pone.0034328

Reply: Thank you for suggesting this reference. Following your advice, we added this reference (Reference 18 in the revised version) to the revised version (Lines 51-53 in therevised version) of our manuscript.

In a review, the readers expect to see more citations of original research papers, instead of just another review. Perphaps, you may wish to cite the following paper which use omics techiniques to identify panels of markers in maternal plasma for prediction of preterm birth:

I am sure there are other original reseach papers which you could cite for Lines 181-183, 293-297. So please cite those as well instead of just reviews.

Reply: Thank you for this suggestion. Yes - we agreed to cite original research papers rather than just review papers. In the revised version (Lines 181-183 → Lines 269-271 in revised version) of our manuscript, we incorporated the references of both original and current research articles instead of review articles (References 126-128 in the revised version), and concerning the omics techniques in the revised version (Lines 293-297 → Lines 349-350 in the revised version) of our manuscript we cited more references regarding the identification of predictive markers for PTB (References 31, 116, 124, and 140 in the revised version).

Thank you,

Sincere regards

Reviewer 2 Report

This review summarizes recent research on the role of the reproductive tract microbiome and its metabolites in relation to preterm birth.  These are topics of significant interest given the clinical significance of preterm birth, and the variation in incidence of prematurity among different populations and geographical locations. 

Major Concerns

The focus on the microbiome metabolome is the strength of the manuscript, but having said that, the review is remarkably deficient in that major contributions to our understanding of the role of the microbiome in the pathophysiology of preterm birth are not discussed/cited (e.g., Fettweis et al. Nature Medicine, 2019, Elovitz et al. Nature Communications, 2019, Gerson et al., AJOG, 2020 among many others), nor are highly relevant recent publications on metabolites in the vagina related to the microbiota (e.g., Srinivasan et al. mBio, 2015; Parolin et al. Frontiers in Microbiology, 2018; Ceccarani et al., Scientific Reports, 2019; Laghi et al. PLoS ONE, 2021; McMillan et al. Scientific Reports, 2015; Chee et al., Microbial Cell Factories, 2020). 

The authors do not discuss the known differences in the vaginal microbiome among different populations/ethnicities and the potential relationships to metabolites and preterm birth.

Significant isolate-specific (strain) differences in vaginal microbe genomes that affect pathogenicity of bacteria thought to play a role in preterm birth like Gardnerella vaginalis have been described.  This important variation is not evident from standard 16sRNA profiling and is not discussed in this review.  Likewise, differences in Lactobacillus crispatus isolates which may influence their protective role against vaginal dysbiosis have been described but are not discussed.

The metabolome of the reproductive tract is complex, and its role in mediating the pathogenic activity of the microbiome involves microbial as well as maternal dietary and metabolic products. Beyond, vaginal epithelial cell glycogen, other maternal metabolites/nutrients have an important role in shaping the vaginal microbiome and its pathogenic potential.  For example, host vitamin D status affects the composition of the vaginal microbiome during pregnancy, and maternal cholesterol metabolism is involved in the activity of microbial toxins (e.g., vaginolysin, a cholesterol dependent cytolysin).   Obesity also alters the vaginal microbiome (Raglan et al., Microbiome, 2021). The complexity of the reproductive tract niche with respect to microbial metabolism is not given adequate consideration.  Additionally, the complexity of metabolite profiles related to differences in vaginal community state types is not considered.

A discussion of approaches to illuminate the metabolic profiles of the vaginal microbiota through shotgun sequencing of microbial DNA, quantitative assessment of microbial gene expression profiles, and targeted and untargeted mass spectrometry metabolomics.

Minor points

Figure 1 presents a hodgepodge of factors/agents that could possibly be involved in the pathophysiology of preterm birth. Unfortunately, it creates confusion rather than clarity, especially because the legend does not provide sufficient detail.  The authors should consider simplifying this figure.

The figures contain multiple colors and the meaning if any of this coding scheme is not defined in the figure legends (e.g., Figure 1, Figure 2, Figure 4).

The manuscript should be purged of broad and simplistic statements (e.g., Page 11, line 266 “A prudent or traditional dietary pattern during pregnancy decreases the risk of PTB…”).

Author Response

Reviewer 2.

Reviewers reply

This review summarizes recent research on the role of the reproductive tract microbiome and its metabolites in relation to preterm birth.  These are topics of significant interest given the clinical significance of preterm birth, and the variation in incidence of prematurity among different populations and geographical locations. 

Reply: Thank you for your insightful review summary and for providing us with valuable comments. Following your major and minor comments, our replies are as follows:

Major Concerns

The focus on the microbiome metabolome is the strength of the manuscript, but having said that, the review is remarkably deficient in that major contributions to our understanding of the role of the microbiome in the pathophysiology of preterm birth are not discussed/cited (e.g., Fettweis et al. Nature Medicine, 2019, Elovitz et al. Nature Communications, 2019, Gerson et al., AJOG, 2020 among many others), nor are highly relevant recent publications on metabolites in the vagina related to the microbiota (e.g., Srinivasan et al. mBio, 2015; Parolin et al. Frontiers in Microbiology, 2018; Ceccarani et al., Scientific Reports, 2019; Laghi et al. PLoS ONE, 2021; McMillan et al. Scientific Reports, 2015; Chee et al., Microbial Cell Factories, 2020). 

Reply: Thank you for your positive view of our manuscript’s strengths. The manuscript focused on the microbiome and their generated metabolites in the pathophysiology of preterm birth. In our revised version we added more discussion regarding the microbiota in the pathophysiology of PTB and microbiota-metabolites in the vagina with suggested references as well as more relevant recent references are cited

  • Normally, the dominance… modulation of host immunity [ 39-43] (Lines 83-89).
  • Additionally, communities…. highest median pH (5.3 ± 0.6) [45,46] (Lines 102-103).
  • The abundance of Lactobacillus spp. … BV and PTB [8, 31] (Lines 121-128).
  • However, one … the pregancises (Figure 2, Table 1) [53,65,66] (Lines 139—147).

Cited suggested references

  • Fettweis et al. Nature Medicine, 2019. Reference 123 in the revised version.
  • Elovitz et al. Nature Communications, 2019, Reference 44 in the revised version.
  • Gerson et al., AJOG, 2020. Reference 47 in the revised version.
  • Srinivasan et al. mBio, 2015. Reference 53 in the revised version.
  • Parolin et al. Frontiers in Microbiology, 2018. Reference 141 in the revised version.
  • Ceccarani et al., Scientific Reports, 2019. Reference 143 in the revised version.
  • Laghi et al. PLoS ONE, 2021. Reference 33 in the revised version.
  • McMillan et al. Scientific Reports, 2015. Reference 124 in the revised version.
  • Chee et al., Microbial Cell Factories, 2020. Reference 46 in the revised version.

The authors do not discuss the known differences in the vaginal microbiome among different populations/ethnicities and the potential relationships to metabolites and preterm birth.

Reply: Thank you for pointing out the importance of the vaginal microbiome among different populations/ethnicity. In our revised version we discussed ethnicity concerning relation to metabolites and preterm birth.

  • … ethinicity (PTB higher in non-Hispanic black women) [7] (Lines 40-41).
  • Instead of defencive… American women [42] (Lines 89-92).
  • A previous study … community state types (CST) [43] (Line 93-97).
  • The microbial dysbiosis … birth outcome [31, 36] (Lines 113-115).

Significant isolate-specific (strain) differences in vaginal microbe genomes that affect pathogenicity of bacteria thought to play a role in preterm birth like Gardnerella vaginalis have been described. This important variation is not evident from standard 16sRNA profiling and is not discussed in this review.  Likewise, differences in Lactobacillus crispatus isolates which may influence their protective role against vaginal dysbiosis have been described but are not discussed.

Reply: Thank you for your insightful suggestion regarding the standard 16S rRNA profiling. Concerning your review regarding important protective and defensive role against vaginal dysbiosis of Lactobacillus spp. (Lines 83-89, Lines 142-147, Lines lines 246-266), and the pathogenic role of Gardnerella vaginalis, Ureaplasma are discussed in the revised version with relevant references (Lines 143-147, Lines 246-266). The dominance-specific strains of Lactobacillus spp. specially Lactobacillus crispatus reflected term birth characterized., while Gardnerella vaginalis commonly isolated from patients with BV. The Lactobacillus spp. characterized by pyrosequencing of barcoded 16S rRNA genes and clustered into five groups called community state types (CST) (Lines 95-108). In case Gardnerella vaginalis characterization by 16S rRNA gene of V2 region, the PCR products as an indicator of BV [47-48] (Lines 108-110).

The metabolome of the reproductive tract is complex, and its role in mediating the pathogenic activity of the microbiome involves microbial as well as maternal dietary and metabolic products. Beyond, vaginal epithelial cell glycogen, other maternal metabolites/nutrients have an important role in shaping the vaginal microbiome and its pathogenic potential.  For example, host vitamin D status affects the composition of the vaginal microbiome during pregnancy, and maternal cholesterol metabolism is involved in the activity of microbial toxins (e.g., vaginolysin, a cholesterol dependent cytolysin).   Obesity also alters the vaginal microbiome (Raglan et al., Microbiome, 2021). The complexity of the reproductive tract niche with respect to microbial metabolism is not given adequate consideration.  Additionally, the complexity of metabolite profiles related to differences in vaginal community state types is not considered.

Reply: Thank you for the insightful comment and suggestions regarding the complexity of microbial pathogenicity, microbiota- metabolites generated by maternal dietary metabolic products, and obesity which have important roles in birth outcomes.

  • Yes, beyond glycogen, other maternal metabolites/nutrients are important in shaping the vaginal environment and its potential pathogenicity in PTB. Higher microbial richness and diversity in different bio-fluids (blood, urine, cervicovaginal fluid, amniotic fluid) directly affect the production of metabolites which affect the normal developmental physiology and birth outcome [31,36] (Lines 111-115). As we found in our recently published article, the consumption of high levels of protein and carbohydrates generate high levels of pathogenic metabolites like Ethanol, Methanol, Formate, TMA, and TMAO in various biot-fluid of PTB women (Lines 230-232, and 291-292, Reference 23 in the revised version). These metabolites are also shown in the figures
  • Vitamins also play a significant role in fetus development and birth outcome. As the reviewer mentioned, vitamin D shows immunomodulatory effects in pregnancy. The existing body of literature reveals that the vitamin A metabolite (retinoic acid) is a key player in embryogenesis. A recently published article of ours explains how we observed the role of vitamin A metabolites (retinyl palmitate, At-Retinal, 13-cis-Retinoic acid) in PTB. We described these issues in our revised version. (Lines 342-348, and Lines 419-420; References 75 in the revised version). We prepared a figure on the basis of our recent findings of vitamin A metabolites associated with PTB. If you agree, we will insert this figure in our manuscript (Figure 5).

  • As per your suggestion, we cited some more references in connection with obesity and microbiota, along with the reference suggested by the reviewer (Raglan et al., Microbiome, 2021 [8]. (Lines 43-44, Lines 161-164 in the revised version). Consumption of a high-fat diet increases chances of obesity and increased maternal cholesterol toxin (Inerolysin, a cholesterol-dependent cytolysin produced by Lactobacillus iners) which increases the vaginal pH and facilitate the PTB (Lines 205-207).

A discussion of approaches to illuminate the metabolic profiles of the vaginal microbiota through shotgun sequencing of microbial DNA, quantitative assessment of microbial gene expression profiles, and targeted and untargeted mass spectrometry metabolomics.

Thank you for suggesting that we add more discussion regarding metabolic profiles of vagina microbiota using modern techniques. In our revised version, we added your suggested contents of short gun sequencing of microbial DNA (Lines 268-270), targeted and untargeted metabolomics (Lines 290-293), polymorphisms or hyper-methylation in genes or RNA transcripts (Lines 51-53), and mass spectrometry metabolomics analysis (Lines266-268, and Lines 313-315).

Minor points

Figure 1 presents a hodgepodge of factors/agents that could possibly be involved in the pathophysiology of preterm birth. Unfortunately, it creates confusion rather than clarity, especially because the legend does not provide sufficient detail.  The authors should consider simplifying this figure.

Reply: Considering your valuable suggestion, we created a simplified picture so that our readers could better understand the pathophysiology of preterm birth. Because our manuscript is focused on microbiota, metabolites, which might induce the inflammatory signals of preterm birth, so we mentioned these factors only in Figure 1. (Line 57), and the environmental and clinical factors we mentioned in the text form (Lines 38-44).

The figures contain multiple colors and the meaning if any of this coding scheme is not defined in the figure legends (e.g., Figure 1, Figure 2, Figure 4).

Reply: Thank you for pointing out this issue, we defined the color coding scheme in our revised version. Yes - the figure contains multiple colors which reflect different meanings which are explained in the figures legends revised version.

  • Figure 1. Microbiota-metabolites and inflammatory markers in preterm birth. O: Oral; G: Gut; B: Blood; U: Urine; V: Vagina; C: Cervix; P: Placenta; A: Amniotic fluid. TMA: Trimethylamine, TMAO: Trimethylamine N-oxide; IL: Interleukin; TNF: Tumor necrosis factor; MMP: Matrix metalloproteinase; CCL: C-C motif chemokine ligand; CXCL: C-X-C motif chemokine; Ptgds: Prostaglandin D2 synthase; Oxtr: Oxytocin receptor; Cx: Connexin; NFkB: Nuclear factor-kappa B; Cox: Cyclooxygenase; PGDH: Prostaglandin dehydrogenase. Dashed lines indicate several steps, ↓decreased, ↑increased level. Location followed by color code (Lines 63-69).
  • Figure 2. Vaginal microbiota Eubiosis and Dysbiosis: Effect of beneficial and pathogenic microbiota in the vaginal environment. Pink = normal, while red = inflammation conditions (Lines 149-150).
  • Figure 3. Generation of microbiota metabolites. The blue color refers to carbohydrates, the red color refers to proteins, the yellow color refers to lipids, and the green color refers to vitamins. From dark color to light color indicate metabolites forms generated by their specific microbiota. From dark to light color sheds indicated derivatives of metabolites conversion (Lines 209-212).
  • Figure 4. Molecular mechanism of microbiota-metabolites in preterm birth. Signaling of PAMPs and DAMPs derived from gut-microbiota and dietary molecules have been activating TLRs in PTB. The blue and gray colors refer to carbohydrates, the red color refers to proteins, the yellow color refers to lipids, and the green color refers to vitamins. From dark color to light color indicate metabolites forms and their generated inflammatory markers (Lines 362—366).

The manuscript should be purged of broad and simplistic statements (e.g., Page 11, line 266 “A prudent or traditional dietary pattern during pregnancy decreases the risk of PTB…”).

Reply: Thank you for your insightful comment. In the revised version of our manuscript, we tried to use simplistic statements and changed the sentence as per the reviewer with relevant references. A healthy dietary pattern containing fibrous food consumption during pregnancy decreases the risk of PTB by increasing beneficial SCFA metabolites and lowering pH compared when compared with a diet consisting of the consumption of western-style junk foods [156,157] (Line 266 → Line 395-397).

Thank you

Sincere regards

Round 2

Reviewer 2 Report

The revisions to the manuscript address all concerns raised in the initial review.  With copy-editing to improve English, this will be a significant contribution to the literature related to the reproductive tract microbiome and preterm birth.